# Assessing Coastal Vulnerability to Storms: A Case Study on the Coast of Thrace, Greece

**Iason A. Chalmoukis**

Department of Civil Engineering, University of Patras, 26500 Patras, Greece; ichalmoukis@upatras.gr

**Abstract:** Climate change is expected to increase the risks of coastal hazards (erosion and inundation). To effectively cope with these emerging problems, littoral countries are advised to assess their coastal vulnerabilities. In this study, coastal vulnerability is first assessed by considering two basic storm-induced phenomena, i.e., erosion and inundation. First, the erosion is computed using the numerical model for Storm-induced BEAch CHange (SBEACH), whereas the inundation is estimated using two different empirical equations for comparison. Then, the integration of the vulnerabilities of both storm-induced impacts associated with the same return period permits the identification of the most hazardous regions. The methodology is applied to the coast of Thrace (Greece). The majority of the coastline is not vulnerable to erosion, except for some steep and narrow beaches and the coast along the city of Alexandroupolis. Beaches with very low heights are highly vulnerable to inundation. Half of the studied coastline is considered highly or very highly vulnerable, whereas the other half is relatively safe. The above results will help decision-makers choose how to invest their resources for preventing damage.

**Keywords:** coastal vulnerability; integrated coastal management; erosion; inundation

## 1. Introduction

Coastal vulnerability is generally defined as the potential of a beach to be damaged by storm-induced phenomena [1]. It is quantified by comparing the magnitude of the impact to the adaptation ability of the coast [2]. The impact is estimated using the intensity of the storm-induced processes, whereas the capacity of the beach to cope with the considered impacts is derived from its geomorphology (i.e., beach slope, width, and height). The most common storm-induced impacts are erosion and inundation [3].

Throughout the last century, pressure on coastal areas has increased due to urbanization and large migration towards them. Approximately, a 70% increase in population has been observed worldwide in low-elevation coastal zones [4]. In particular, the population around the Mediterranean Sea is estimated to reach 572 million by 2030 [5]. Furthermore, climate change is expected to have long-term impacts with frequent and intense extreme storm events and a permanent 1.5 °C increase in global surface temperature by 2050 [6]. Consequently, sea level rise will threaten more coasts, and in combination with storm events and local erosion trends, this can impose severe flood risks. Due to the above, the Mediterranean coastline is identified by the Intergovernmental Panel on Climate Change as a vulnerable zone with a high risk of inundation, coastal erosion, and, in general, land degradation [6]. The increasing erosion and flood phenomena arising in the Mediterranean push public administrations towards a strategic approach for integrated coastal zone management (ICZM) with an emphasis on coastal protection. More than a decade ago, the importance of including hazard assessments in coastal zone policies was highlighted by the Protocol on ICZM in the Mediterranean [7]. It recommends that its littoral countries address the effects of natural disasters along their coastlines by assessing their vulnerability.

Following this recommendation, Mendoza and Jimenez [1] estimated the erosion potential of the Catalonian coast in Spain using the Storm-induced BEAch CHange (SBEACH)

numerical model. From the results of the beach retreat, a five-class storm categorization was proposed for the Catalonian coast. Monioudi et al. [8] assessed the erosion risk of the eastern Cretan coastline in Greece under sea level rise. Three analytical equations were used to estimate the long-term beach evolution, whereas short-term erosion was modeled using three numerical response models (XBeach, Leont'yev, SBEACH). They found that the last two models presented similar results. Furthermore, an evaluation of the economic impact of erosion on tourism revenue for Crete Island in Greece was presented by Alexandrakis et al. [9], following a similar approach to that of McLaughlin et al. [10]. In contrast, De Leo et al. [11] studied only the flooding potential of Lalzit Bay in Albania using offshore wave climate (regional) and nearshore wave climate (local). Their analysis concluded that the results vary in the run-up estimation, and thus the coastal vulnerability, due to the different wave climates.

Studies that considered both the erosion and inundation impacts to assess the integrated coastal vulnerability have also been performed [3,12–15]. The advantage of these articles is that the vulnerability to erosion and inundation were evaluated separately, and then the contribution of the forcing (storm properties) and receptor (geomorphology) were quantified for the overall vulnerability. A disadvantage in the work of Jiménez et al. [12] is that they did not provide a method to calculate the return period and thus it was arbitrarily selected. The drawback in the study of Bosom and Jiménez [3] is that an empirical formula specifically derived from numerical simulations of SBEACH along the Catalonian coast [1] was used to predict the erosion. The results showed that beach erosion estimations similar to the predictions of SBEACH can be obtained in a simple manner for Catalonian beaches using the storm characteristics and the proposed erosion classes. However, the empirical formula cannot be used for other coasts without appropriate calibration. Ferreira et al. [15] analyzed several European coasts, with three of them being in the Mediterranean Sea. The numerical model XBeach was used to estimate the erosion, and XBeach coupled with the overland flood model LISFLOOD-FP were used to compute the inundation. The advantage of this approach is that their results are more reliable because the impacts are analyzed in detail. However, its drawback is the increased computational time, since setting up XBeach and LISFLOOD-FP requires more workload.

It should be mentioned that, in addition to the Mediterranean Sea, similar studies have also been performed worldwide (e.g., Brazil [16], Bangladesh [17], Colombia [18], and South Korea [19]). A GIS-based coastal vulnerability assessment of state of Pará, Brazil, was performed by Szlafsztei and Sterr [16]. A new approach implementing fuzzy logic based on geospatial techniques was used to analyze the vulnerability of the coast of Bangladesh [17]. The authors claim that they can analyze the vulnerability at a cell size of 10 m. The drawback of this approach is the reliability and access of several necessary GIS data. The vulnerability of the Caribbean and Pacific coast of Colombia was estimated roughly through a semi-quantitative approximation by applying relative indices to different variables [18]. The advantage of this approach is that they included human aspects in their variables, such as housing density, contamination, etc.

The aim of this article is to present a tool to assess the integrated coastal vulnerability to storm-induced inundation and erosion following a similar probabilistic approach as in the work of Bosom and Jiménez [3] and apply it on the coast of Thrace, Greece (Figure 1). The modification of the present study is the use of SBEACH to compute the beach retreat, instead of an empirical equation. In addition, two different empirical formulas to estimate the wave run-up are used to analyze their differences, and the inundation vulnerability framework includes overtopping to better determine the flooding impact. After calculating the time series of the two hazards, extreme probability functions are fitted to estimate their magnitudes with large return periods. When both probability distributions are known, an accepted risk and a time period of concern are selected to estimate a specific return period. This permits the comparison of vulnerabilities with the same return period and the identification of the most vulnerable areas. Finally, a simple approach to integrate the vulnerabilities of both hazards is presented.

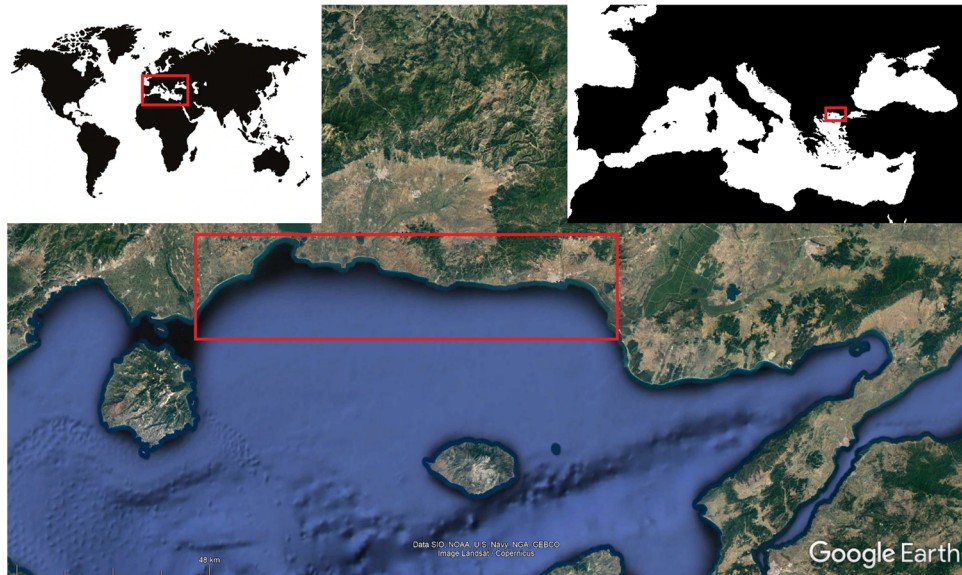

**Figure 1.** Panoramic view of the globe (**top-left** corner) with indication of the Mediterranean basin. The red rectangle at the Mediterranean map (**top-right** corner) indicates the studied area, coast of Thrace, Greece (Photo courtesy of Google Earth).

## 2. Study Coast and Data

The studied coastline is in the Mediterranean Sea and, particularly, in the region of Thrace, northeast Aegean Sea of Greece (Figure 1). Its length is about 108 km, and its boundaries are the Nestos and Evros rivers, which are located at the west and east, respectively. Along the river streams, three dams were constructed, resulting in the reduction of the sediment towards the coast [20]. Apart from the two large rivers, it comprises of long straight sandy beaches, cliffs, pocket beaches, deltas, lagoons, salt pits and a lake. The only city on the coastline is Alexandroupolis. The coastal zone of this city is of high economic value due to its port where commercial, transportation and tourist activities occur. In contrast, the lagoons, the lake, and the deltas have low economic value, because any investment will face significant environmental restrictions due to the Ramsar and Natura 2000 protection [21,22]. The socio-economic structure of the rest of the coastline is based on tourism, aquaculture, agriculture and residential developments, and its economic value is characterized as medium.

The coastline experiences mild wave climate, which is typical at the North-Eastern Mediterranean (microtidal and fetch-limited wave conditions) [23]. To characterize the storms, 3 h hindcasts of the WAM model for ten years (January 1995–December 2004) at four stations (Figure 2) were used, provided by the Hellenic Centre of Maritime Research. The wave dataset included the significant wave height, $H_s$, wave peak period, $T_p$, wave direction, $H_{dir}$, wind speed, $W_s$, and wind direction, $W_{dir}$. It should be stressed that the Aegean Sea presents short fetches and storms of limited duration, which may lead to model errors in comparison to ocean predictions [24]. However, the hindcasted values are considered representative of the actual wave conditions because they were compared to available measurements from buoys, and no significant differences were observed [25].

Information about the basic beach dimensions (height, width, and slope) was obtained from the Hellenic Military Geographical Service through a digital terrain model with 5 m resolution. The beach slope is mild (~0.002) around the deltas and steep (~0.067) at the center of the coastline. The sediment diameter varies between 0.22 mm and 0.5 mm [20]. The selected beaches for the present vulnerability assessment are shown in Figure 3.

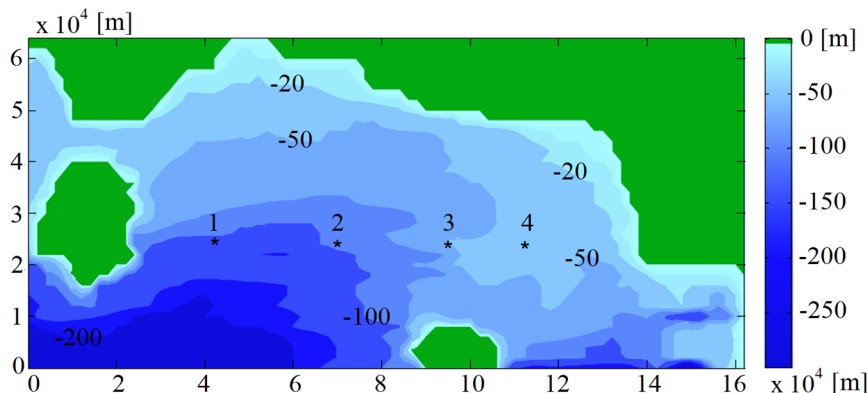

**Figure 2.** The bathymetry of the studied area with indication of the four stations (1, 2, 3, 4). Thasos and Samothraki islands can be observed at the west and south, respectively.

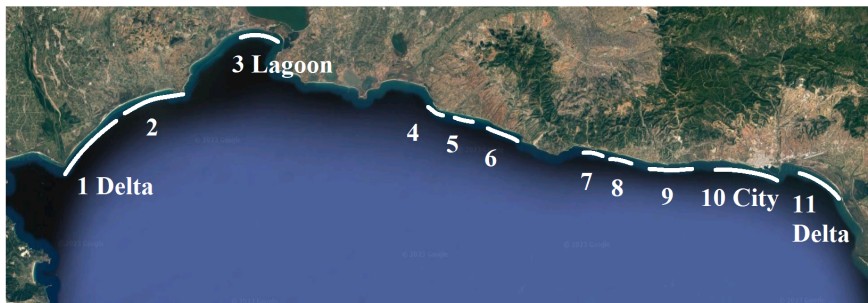

**Figure 3.** The studied coastline with the eleven selected beaches.

The first and last beach (1 and 11) are located at the deltas of the two rivers, Nestos and Evros, respectively. Area 3 (Lagoon) is a barrier beach, which is a narrow strip of land with low elevation. It is a crucial buffer that protects the lagoon against storm damage and flooding, but its existence is very fragile. The rest of the studied beaches (2 until 10) are typical sandy ones, whereas beach 10 (City) runs along the city of Alexandroupolis. The characteristics of each beach are presented on Table 1.

**Table 1.** Characteristics of the studied beaches.

|  | Length [m] | Width [m] | Height [m] | Slope | Buoys |
|---|---|---|---|---|---|
| 1 Delta | 2000 | 500 | 0.9 | 0.002 | 1 |
| 2 | 1300 | 30 | 0.4 | 0.013 | 1 |
| 3 Lagoon | 700 | 20 | 0.3 | 0.018 | 1 |
| 4 | 300 | 20 | 0.4 | 0.020 | 2 |
| 5 | 500 | 20 | 0.8 | 0.040 | 2 |
| 6 | 1500 | 15 | 1 | 0.067 | 3 |
| 7 | 1000 | 30 | 2 | 0.067 | 3 |
| 8 | 1100 | 30 | 2 | 0.067 | 3 |
| 9 | 6000 | 25 | 0.8 | 0.032 | 3 |
| 10 City | 9000 | 15 | 0.5 | 0.033 | 4 |
| 11 Delta | 8000 | 125 | 1 | 0.008 | 4 |

## 3. Methodology

The methodological framework to assess integrated coastal vulnerability to storm-induced impacts is schematized in Figure 4. It consists of four main steps: (a) estimation of the storm-induced phenomena, (b) calculation of the probability distribution of the induced hazards, (c), selection of the return period, and (d) integration of the coastal vulnerability.

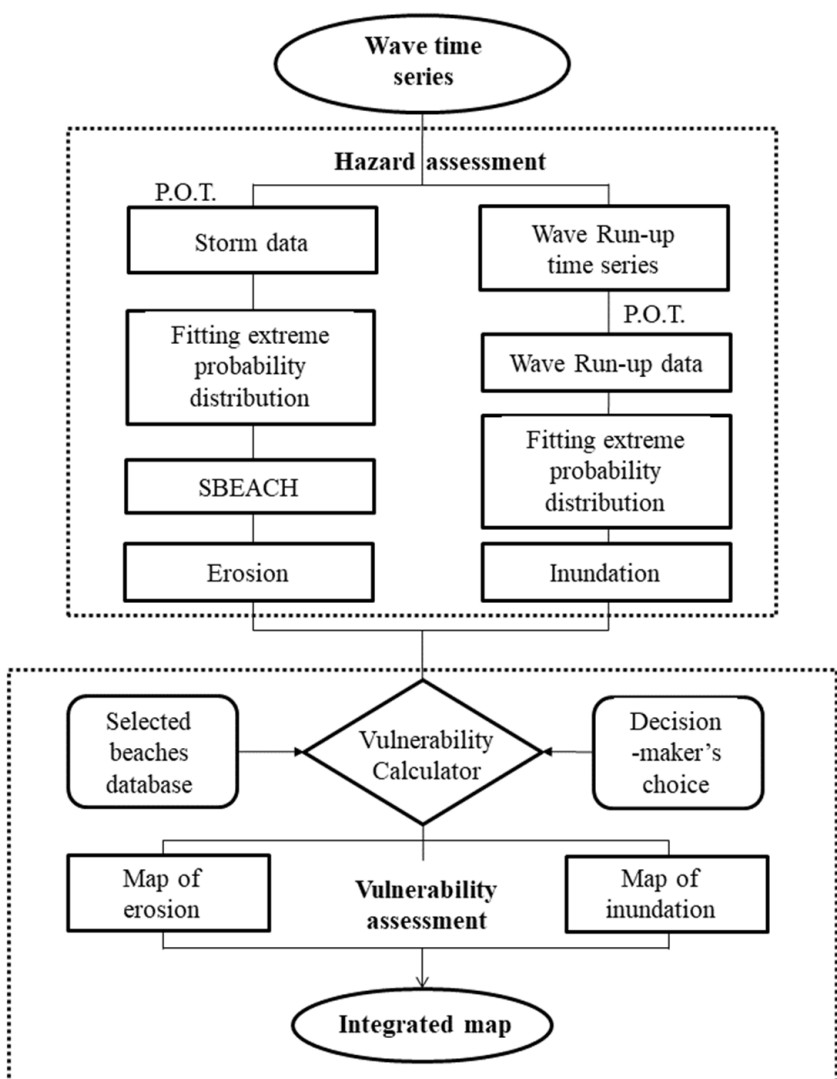

**Figure 4.** Methodological framework for integrated coastal vulnerability assessment to storms.

The event approach [26] is followed to estimate the erosion, since it is impossible to calculate its time-series, but only the erosion produced by independent storms. Therefore, wave height extreme distributions are first calculated from the storm time-series, then, other related parameters (i.e., wave period and storm duration) are estimated, and finally, the storm-induced erosion for each return period is computed.

On the other hand, the response approach [26] is used to estimate the inundation, because the run-up time-series can be directly calculated from wave data. Afterwards, extreme probability distributions are fitted, and from them, the inundation probability distribution is estimated. This reduces the uncertainty in the present analysis, since it allows the hazard magnitude associated with a given probability of occurrence to be obtained without assuming relationships between the driving variables [15].

Once the erosion and inundation probability distributions are estimated, vulnerability maps are drawn based on a selected return period. Following this methodology, decision-makers (public or private administrators, municipalities, coastal managers, etc.) can select an acceptable return period for each area of the coastline depending on its characteristics (economy, infrastructure, environment, etc.). Selecting spatially varying return periods allows comparisons of vulnerabilities associated with different probabilities.

*3.1. Erosion Parameterization*

Cross-shore erosion occurs mainly during extreme storms, and unlike long-shore erosion, which is difficult to observe in the short-term, it can transport large sediment volumes over intervals as short as one day. Since storm-induced hazards are considered in this project, only cross-shore erosion will be estimated.

To start with its vulnerability assessment, first step consists of finding the hazards forcing, i.e., identify storms in the study area. Subsequently, in order to do so, a storm data set must be built using the existing wave time-series. In this study, the peak-over-threshold (P.O.T.) method was used, where a storm is defined as an uninterrupted sequence of waves with height exceeding a threshold value and direction towards the under-consideration beaches. The wave height threshold is selected by calculating the 95% quantile of its cumulative distribution function. Then, extreme probability (Gumbel, Generalized Extreme Value, Generalized Pareto, and Weibull) distributions were used to fit the storm wave data. Apart from the extreme wave height, its corresponding period and storm duration are computed using regression analysis to the existing wave data. With these parameters, the erosion of every storm can be predicted. Finally, the numerical model SBEACH is selected to estimate the beach profile response. A detailed description of the model is available in [27,28], whereas its disadvantages were presented by Thieler et al. [29]. The direction and rate of cross-shore sediment transport predicted by SBEACH is based on empirical criteria derived from wave-tank experiments. Its fundamental assumption is that the beach is changed only by cross-shore erosion, resulting in a redistribution of sediment across its profile with no net gain or loss of material (sand conservation). Furthermore, it should not be used to examine profile changes near jetties or similar structures because, in these cases, profile changes might be controlled more by the interruption of long-shore transport than by cross-shore. Therefore, SBEACH should only be applied if long-shore erosion can be neglected, as is in this case. In addition, it is an easy and fast numerical model that produces reliable results [8].

In this study, beach erosion is characterized by the beach retreat, $\Delta X$, that is defined as the shoreward displacement of the initial beach line after a storm (Figure 5). This is a simplification of a real beach response because pre-storm morphology, among other factors, affects the induced erosion [30]. Furthermore, event grouping, when significantly more erosion, which can occur from two consecutive storms [31], is not considered. However, the objective is not to reproduce the exact beach response to storms but to estimate an order of magnitude of the erosion.

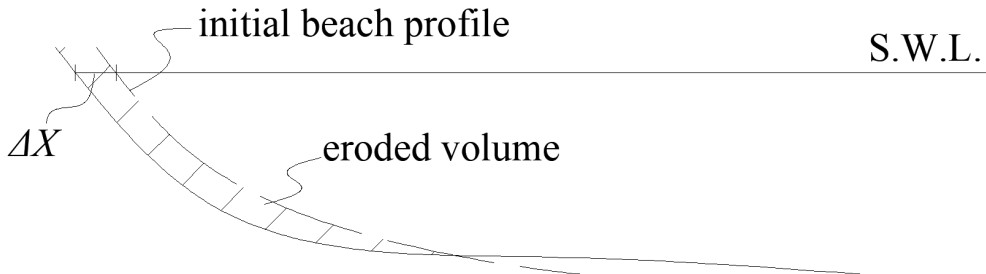

**Figure 5.** Initial and final beach profile. The eroded volume and beach retreat are marked. S.W.L. is defined as standing water level.

The final step includes the ability of a coast to cope with erosion in order to quantify its vulnerability. The parameter used as a characteristic of the coast resilience is the beach width, $W$. Wide beaches do not face significant problems even if their short-term erosion is large. Therefore, the lowest vulnerability will occur when beach width is greater or equal to the minimum required beach width, $W_{min}$, plus the beach retreat, $\Delta x$:

$$W \geq W_{min} + \Delta x \qquad (1)$$

The minimum required beach width is a distance to maintain a coast operative and to avoid direct exposure of the hinterland to seawater [3]. In this work, $W_{min}$ = 10 m to let machinery work operate after storm damages. On the other hand, the highest vulnerability is defined when beach width is less or equal to its retreat:

$$W \leq \Delta x \tag{2}$$

This is the case when a beach is fully eroded, and the infrastructure is exposed to waves. The erosion vulnerability is quantified using a linear function between Equation (1), lowest value (safe beach), and Equation (2) highest value (vulnerable beach). It is also divided in four qualitative classes (green: lowest, yellow: low, orange: high, and red: highest).

### 3.2. Inundation Parameterization

In general, inundation is caused by a combination of high-water levels due to tide and wave run-up. Consequently, joint probability analysis should be performed to estimate the highest water level during a storm. However, the tidal range in the studied coastline is very low [23], and thus, it can be neglected. The wave run-up describes the phenomenon when an incoming wave climbs the beach profile up to a level that can be higher than its crest. The vertical distance between the S.W.L. and the highest point reached by the 2% of the incident waves is called run-up, $R_{2\%}$. For this study, the wave run-up time series is computed using the formula [32]:

$$R_{2\%} = 1.1 \left( 0.35 \tan \beta (H_s L_o)^{0.5} + \frac{[(H_s L_o (0.563 \tan \beta^2 + 0.004)]^{0.5}}{2} \right) \tag{3}$$

and the existing wave time series. In Equation (3), $L_o$ (= $1.562 \cdot T_p^2$) is the deep-water wave-length associated with the wave peak period, $T_p$, and $\tan\beta$ is the beach slope. This equation (henceforth, Stockdon eq.) is selected because it was derived from field measurements on natural sandy beaches [32] and it is widely used [3]. However, it was reported that Stockdon eq. might underpredict the wave run-up [33]. Therefore, the equation presented by Reis et al. [34] (henceforth, Reis eq.) is also used to estimate the wave run-up time series and to compare the two empirical formulas,

$$R_{2\%} = \left( \frac{(0.38 + 1.67\xi)H_s}{1.085} \right) \tag{4}$$

The Irribaren number is defined as $\xi = \tan\beta / (H_s / L_o)^{0.5}$. It should be noted that other formulas [35] to estimate the wave run-up could be used. Nevertheless, to increase the reliability of a vulnerability assessment, it is recommended to verify that the selected equations capture the actual processes. The threshold of the wave run-up time series is selected by finding the 95% quantile of the corresponding cumulative distribution functions. After applying the P.O.T. method to the wave run-up time series, extreme distributions are fitted to the highest run-up values.

The beaches resilience against flooding is a function of their height. Higher beaches will be less inundated. Based on these, the lowest vulnerability is defined when beach height, $B$, is greater or equal to the run-up:

$$B \geq R_{2\%} \tag{5}$$

On the other hand, the highest vulnerability is defined when the run-up exceeds beach height by a value $Z$ or more:

$$B \leq R_{2\%} - Z \tag{6}$$

The variable $Z$ needs to be adapted based on the specific conditions of the studied area. For the highest vulnerability, it represents overtopping conditions with significant

water volumes flowing to the hinterland. It can be defined after comparing the average overtopping discharge of every beach with values allowed in order to have a littoral road, pedestrian sidewalk, etc. without any obstructions [36]. In this study, the overtopping discharge, *Q*, was calculated by [33]:

$$Q = A\sqrt{gR_{2\%}^3}\left(1 - \frac{B}{R_{2\%}}\right)^C, \text{ for } R_{2\%} > B \tag{7}$$

where *g* is the gravitational acceleration, *A* = 0.033, and *C* = 10.2 − 0.275/tan*β*. Similarly with erosion, four levels of vulnerability exist, and a linear relationship is assumed between Equations (5) and (6).

*3.3. Integrating Coastal Vulnerability*

After the estimation of the extreme distributions, the last steps of the methodology are to consider an appropriate return period and to integrate the vulnerability against the hazards of inundation and erosion. The return period is estimated by [37]:

$$T_r = \frac{1}{1 - (1 - P)^{\frac{1}{N}}} \tag{8}$$

where *P* is a probability of occurrence, and *N* is a time period of concern. In general, these variables should be defined by the decision-makers of each region, and hence, different return periods can be analyzed. This permits to study different safety levels, which are a function of the hinterland importance. The integration of the vulnerability of both hazards is performed following Table 2. This approach is conservative, because a high or very high vulnerability has more weight for the estimation of the integrated vulnerability.

**Table 2.** Approach to integrate the vulnerability of erosion and inundation. The four different colors classify the vulnerability (green: lowest, yellow: low, orange: high, and red: highest vulnerability).

| | | Erosion | | | |
|---|---|---|---|---|---|
| | | very Low | low | high | very high |
| **Inundation** | very low | very low | low | high | high |
| | low | low | low | high | high |
| | high | high | high | high | very high |
| | very high | high | high | very high | very high |

## 4. Results

*4.1. Erosion Results*

In this study, the threshold value of the wave height to define a storm was found to be 1.5 m. After applying the P.O.T. method in the wave time series, the storm wave data for beach 10 (City) are shown in Figure 6 (left). Among the extreme probability distributions that were used, the Gumbel distribution was chosen (Figure 6-right) as it presented the best fit. The method that was used to calculate its parameters was the maximum product of spacings with 95% confidence intervals. It should be stressed that only extreme wave heights with return periods of maximum 30 years will be considered reliable, because the extent of a time-series must be at least one-third of the duration to which a variable is being extrapolated [38]. For the present case, thirty years are considered sufficient to produce long-term coastal planning.

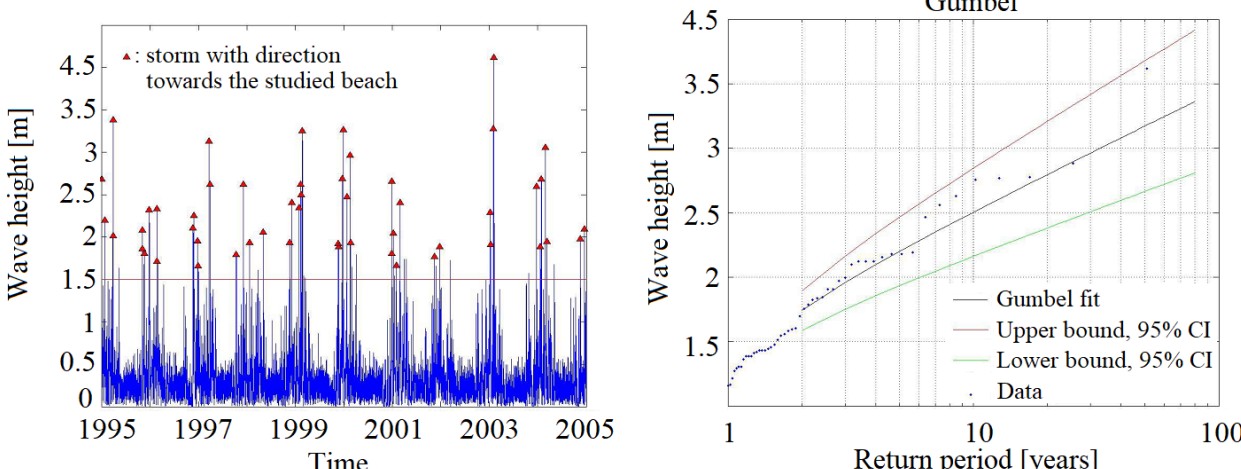

**Figure 6.** **Left**: Wave time series for area 10 (City). The marked peaks above the red line (threshold = 1.5 m) are the storms with a direction towards the studied beach. **Right**: Wave height Gumbel distribution for area 10 (City). The red and green lines are the 95% confidence intervals.

Using the wave height with a return period of 30 years from Figure 6 (right), the final profile of a representative cross-section at beach 10 (City) can be simulated from SBEACH (Figure 7).

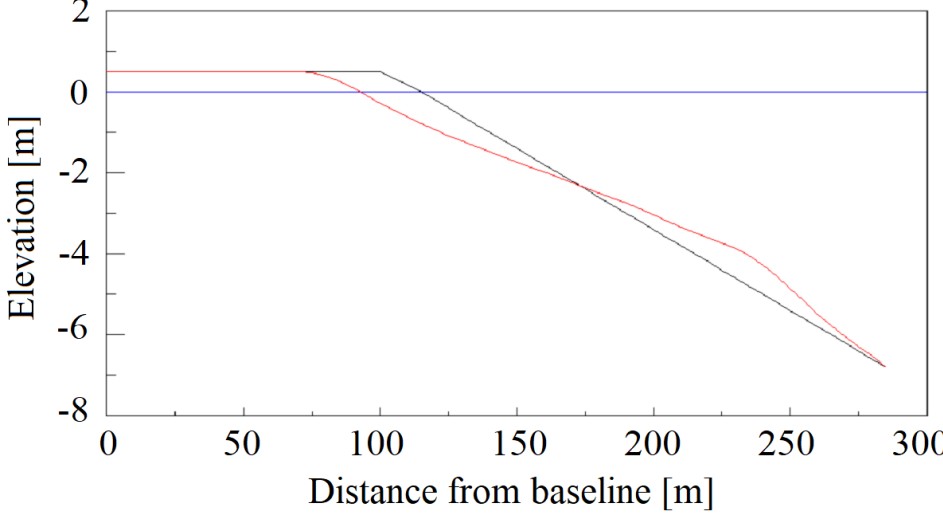

**Figure 7.** Initial (black line) and post−storm (red line) profile of beach 10 (City) for $T_r$ = 30 years.

After following the above procedure for all beaches, the erosion results for three different return periods are summarized in Table 3. It should be noted that beaches 1–4 and 11 are not eroded (even for storms with $T_r$ = 30 years), whereas the others lose much sediment and width. Another interesting fact is that the eroded beaches are located in the sheltered region of Samothraki Island (Figures 2 and 3), receiving storms with lower energy than the non-eroded beaches that are more exposed. This is explained by the beach slope. The mild slope beaches at the deltas and the lagoon (Table 1) dissipate more wave energy, and thus, they are safer against erosion.

**Table 3.** Beach retreat and difference between beach width and retreat for all examined areas with three return periods $T_r$. The four different colors classify the erosion vulnerability (green: lowest, yellow: low, orange: high, and red: highest vulnerability).

| | Beach Width [m] | Beach Retreat [m] $Tr$ [yrs] | | | Beach Width–Beach Retreat [m] $Tr$ [yrs] | | |
|---|---|---|---|---|---|---|---|
| | | 5 | 10 | 30 | 5 | 10 | 30 |
| 1 Delta | 500 | 0.0 | 0.0 | 0.0 | 500.0 | 500.0 | 500.0 |
| 2 | 30 | 0.0 | 0.0 | 0.0 | 30.0 | 30.0 | 30.0 |
| 3 Lagoon | 20 | 0.0 | 0.0 | 0.0 | 20.0 | 20.0 | 20.0 |
| 4 | 20 | 0.0 | 0.0 | 0.0 | 20.0 | 20.0 | 20.0 |
| 5 | 20 | 13.6 | 16.2 | 20.1 | 6.4 | 3.8 | −0.1 |
| 6 | 15 | 16.5 | 18.6 | 22.2 | −1.5 | −3.6 | −7.2 |
| 7 | 30 | 15.9 | 16.1 | 18.8 | 14.1 | 13.9 | 11.2 |
| 8 | 30 | 15.9 | 16.1 | 18.8 | 14.1 | 13.9 | 11.2 |
| 9 | 25 | 6.4 | 7.9 | 9.3 | 18.6 | 17.1 | 15.7 |
| 10 City | 15 | 12.2 | 14.4 | 18.1 | 2.8 | 0.6 | −3.1 |
| 11 Delta | 125 | 0.0 | 0.0 | 0.0 | 0.0 | 0.0 | 0.0 |

The vulnerability map to erosion of the coastline is presented in Figure 8 for the return period of 30 years. Narrow beaches (5, 6 and 10) are those that have erosion problems, whereas beaches 7 and 8 with 30 m width and large beach retreats (18.8 m for $T_r$ = 30 yrs), are not even characterized with low vulnerability. In conclusion, the majority of the coastline is not vulnerable to erosion, except for the steep and narrow beaches 5 and 6, with 20 m maximum width, and the coast along the city of Alexandroupolis (10).

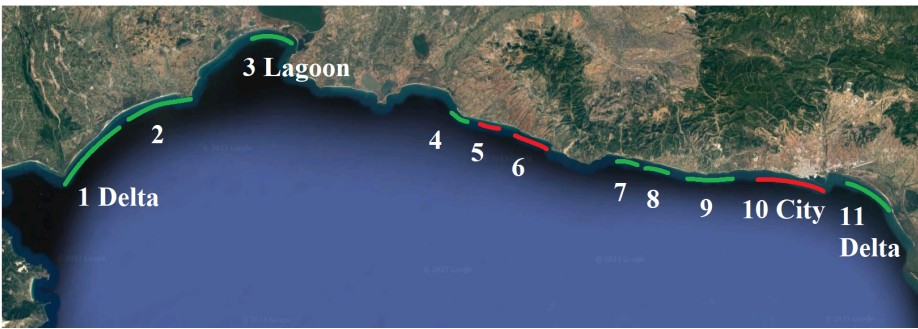

**Figure 8.** Coastal vulnerability map to erosion with $T_r$ = 30 years.

*4.2. Inundation Results*

As it was explained in Section 3.2, the wave run-up time series can be calculated from Equations (3) and (4). In Figure 9, the two different run-up time series due to Equations (3) and (4) are presented for beach 10 (City).

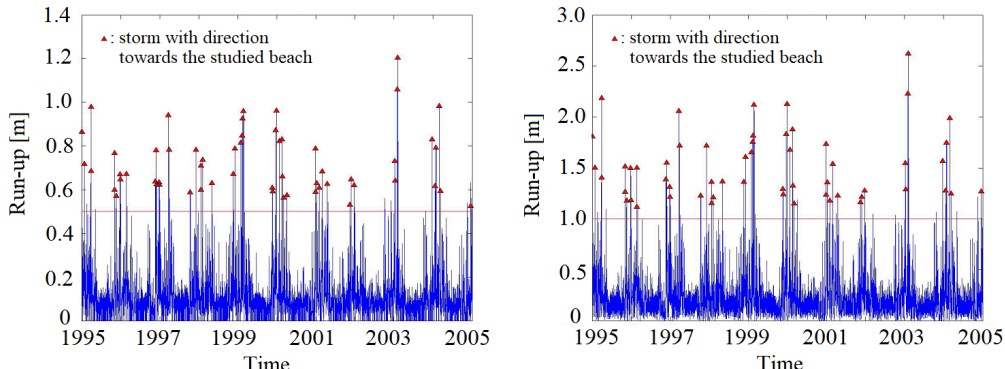

**Figure 9.** Wave run-up time series (**left**—Stockdon eq., **right**—Reis eq.) for beach 10 (City). The red line is the threshold. The marked peaks are the storms with direction towards the studied coastline.

For both run-up time series, the GUMBEL distribution presented the best fit. Hence, it was selected to estimate the wave run-up values for different return periods (Figure 10).

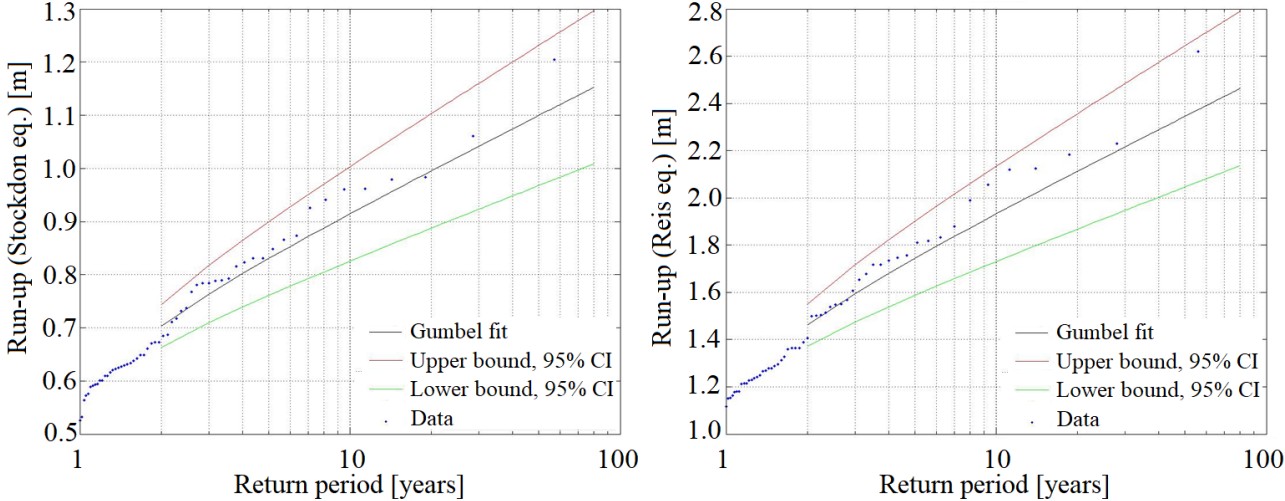

**Figure 10.** Gumbel distribution of wave run-up for beach 10 (City), calculated with Stockdon (**left**) and Reis (**right**) equation. The red and green lines are the 95% confidence intervals.

The inundation results for all examined beaches are presented in Table 4. The wave run-up values calculated by Stockdon eq. are approximately half from the ones calculated by Reis eq. This stresses that using different equations can lead to highly expensive measures against an overestimated hazard, or, in contrast, can pose danger on the under-consideration coastline if the hazard is underestimated. Therefore, selecting the correct equation to describe a physical phenomenon is of paramount importance.

**Table 4.** Beach height and wave run-up predicted by Stockdon and Reis equations for all examined areas with three return periods $T_r$.

|  | Beach Height [m] | Stockdon Eq. [m] $Tr$ [yrs] | | | Reis Eq. [m] $Tr$ [yrs] | | |
|---|---|---|---|---|---|---|---|
|  |  | 5 | 10 | 30 | 5 | 10 | 30 |
| 1 Delta | 0.9 | 0.73 | 0.80 | 0.90 | 1.50 | 1.65 | 1.82 |
| 2 | 0.4 | 0.80 | 0.88 | 1.05 | 1.75 | 1.95 | 2.25 |
| 3 Lagoon | 0.3 | 0.77 | 0.85 | 0.91 | 1.80 | 2.05 | 2.35 |
| 4 | 0.4 | 0.82 | 0.90 | 1.03 | 1.80 | 2.10 | 2.45 |
| 5 | 0.8 | 0.90 | 1.03 | 1.20 | 2.05 | 2.40 | 2.70 |
| 6 | 1.0 | 0.85 | 0.93 | 1.05 | 1.80 | 2.05 | 2.45 |
| 7 | 2.0 | 0.85 | 0.93 | 1.05 | 1.80 | 2.05 | 2.45 |
| 8 | 2.0 | 0.85 | 0.93 | 1.05 | 1.80 | 2.05 | 2.45 |
| 9 | 0.8 | 0.71 | 0.77 | 0.85 | 1.55 | 1.65 | 1.80 |
| 10 City | 0.8 | 0.83 | 0.91 | 1.05 | 1.70 | 1.90 | 2.20 |
| 11 Delta | 1.0 | 0.74 | 0.80 | 0.88 | 1.45 | 1.60 | 1.80 |

It should be noted that the wave run-up values of the lagoon area (3) and the coast of Alexandroupolis (10) are almost the same (Table 4), even though the lagoon (3) is an exposed beach, and the coast in (10) is semi-protected due to the sheltering effect of the islands. This is explained because the wave characteristics are not the only inputs to estimate the run-up; beach slope is also significant. Milder slopes reduce wave run-up, since $tan\beta$ is proportional to $R_{2\%}$ in Equations (3) and (4). Furthermore, large run-up values do not necessarily imply high vulnerability. For example, Reis eq. yields the same run-up for beach 3 (Lagoon) and beach 7 for $T_r = 5$ years (Table 4). However, beach 7 does not have

any flooding problem, because its beach height = 2.0 m > 1.8 m, whereas beach 3 (Lagoon) is completely inundated (beach height = 0.3 m < 1.8 m).

The results of the overtopping discharge using Reis eq. for a return period of 30 years are presented in Table 5. The Reis eq. and 30 years return period were selected to be on the conservative side. It is reported that an overtopping discharge of more than 150 lt/sec causes damage to structures (buildings, revetments, etc.) and it is dangerous and unsafe for pedestrians and driving on coastal roads [35]. Based on this and the corresponding results, the variable $Z$ was set equal to 2 m in Equation (6), e.g., both beach 3 (Lagoon) and 4 experience an overtopping discharge of approximately 175 lt/sec, while having a wave run-up two meters higher than their height.

**Table 5.** Beach height, wave run-up based on Reis eq., and overtopping discharge for all examined beaches.

|  | Beach Height [m] | Reis Eq. [m] $T_r = 30$ yrs | Q [lt/s] $T_r = 30$ yrs |
|---|---|---|---|
| 1 Delta | 0.9 | 1.82 | 10.28 |
| 2 | 0.4 | 2.25 | 139.02 |
| 3 Lagoon | 0.3 | 2.35 | 174.49 |
| 4 | 0.4 | 2.45 | 171.50 |
| 5 | 0.8 | 2.70 | 87.93 |
| 6 | 1.0 | 2.45 | 16.38 |
| 7 | 2.0 | 2.45 | 0.01 |
| 8 | 2.0 | 2.45 | 0.01 |
| 9 | 0.8 | 1.80 | 15.76 |
| 10 City | 0.8 | 2.20 | 100.39 |
| 11 Delta | 1.0 | 1.80 | 5.52 |

Based on the above, the vulnerability maps to inundation are presented in Figures 11 and 12 for Stockdon and Reis equations, respectively. Beaches with very low heights (2, 3, 4, 5 and 10 in Table 4) are highly vulnerable to inundation, whereas beaches 7 and 8, that are two meters high, are not expected to have flooding problems.

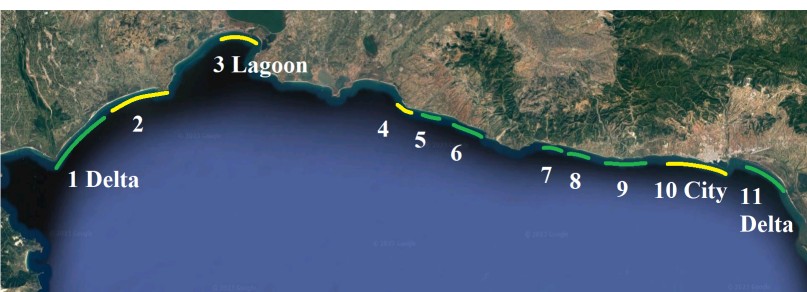

**Figure 11.** Coastal vulnerability map to inundation predicted by Stockdon eq. with $T_r = 30$ years.

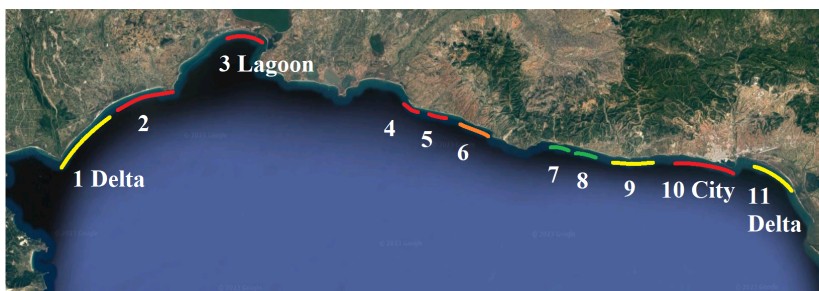

**Figure 12.** Coastal vulnerability map to inundation predicted by Reis eq. with $T_r = 30$ years.

*4.3. Integrated Coastal Vulnerability Assessment*

In this study, the minimum period of concern for coastal protection is set to $n = 10$ years, following the recommendation of the Greek Ministry of Public Works (GMPW). In addition, for the majority of the studied coastline, failure of a beach will not cause human losses and it will have medium economic impact. This corresponds to an accepted probability, $p = 0.3$ (GMPW). By substituting this value in Equation (8), the return period equals to thirty years. Based on this, the vulnerability maps to erosion and inundation (Figures 8, 11 and 12) were presented with a return period of 30 years.

For the integrated coastal vulnerability assessment, the results of Reis eq. were used. This was conducted because the hinterland of this coastline is known to be prone to inundation and Reis eq. yields larger wave run-up values that seem to agree with observations. As a result, the integrated coastal vulnerability to inundation and erosion associated with thirty years return period along the coast of Thrace is presented in Figure 13. From the integrated vulnerability map, half of the studied coastline is considered highly or very highly vulnerable, whereas the other half is relatively safe for $Tr = 30$ years. The barrier beach (3 Lagoon) is characterized by high vulnerability only due to inundation (Figure 12), whereas the coast of Alexandroupolis (10 City) is very highly vulnerable to both inundation and erosion (see also Figures 8 and 12).

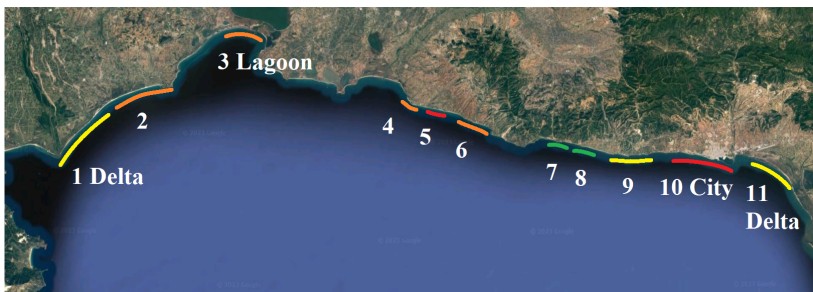

**Figure 13.** Integrated coastal vulnerability map associated with a return period of 30 years.

## 5. Discussion

An assumption to apply the above methodology is that the beach dimensions were averaged for each studied area. This may underestimate or overestimate the vulnerability in some parts. To improve the present assessment in the future, in situ analytical measurements of beach slope, width and height are required. Another point to be considered is that coastal geomorphology needs to be updated as beach profiles evolve constantly. As it was previously mentioned, the hazard intensity and the beach capacity to cope with it depend on the pre-storm morphology. Hence, if decision-makers want to have a reliable live estimation, the present method needs to be complemented with a coastal monitoring plan.

In addition, the present analysis revealed the importance of correctly estimating a hazard in order to have a reliable vulnerability map. Therefore, the used equations and models should be validated in order to increase the reliability of the coastal vulnerability assessments. Most models assume a smooth profile or straight and parallel seabed contours. This is an assumption that may be completely invalid on coasts that have submerged rocks or boulders. In the real world, there is also substantial regional and local variability in grain size. The above should be taken into consideration for the validity of the results.

All in all, the results based on the above hypotheses may contain a degree of uncertainty for some parts of the under-consideration coastline. However, the aim of the present study is not to replace a detailed beach modeling analysis, but to provide a first approach to the vulnerability assessment of a large coastline. This can help decision-makers choose how to invest their resources for preventing damages.

## 6. Conclusions

In this article, the probabilistic vulnerability to storms of the coast of Thrace, Greece was assessed. To this end, the methodology presented by Bosom and Jiménez [3] has

been applied with three modifications. The introduced modifications were the use of the numerical model SBEACH to estimate the beach retreat, instead of an empirical formula. Furthermore, two different run-up equations, instead of one, were used to compare their different estimations concerning the flooding results. Finally, the vulnerability to inundation was defined after considering the corresponding overtopping discharge.

A large vulnerability variation along the coast of Thrace was found, stressing the importance of waves and beach geomorphology to the integrated hazards assessment. The majority of the coastline is not vulnerable to erosion, except for some steep and narrow beaches, and the coast along the city of Alexandroupolis. Beaches with very low heights are highly vulnerable to inundation, whereas beaches that are two meters high, are not expected to have flooding problems. Half of the studied coastline is considered highly or very highly vulnerable, whereas the other half is relatively safe for $Tr = 30$ years. It was also found that an area that is affected by a high-intensity hazard is not necessarily vulnerable, because coastal vulnerability is defined by the beach capacity to cope with an extreme hazard.

This methodology is simple, and it can be similarly applied to other Mediterranean coasts. For different regions, (e.g., beaches with high tide), or under climate change scenarios (e.g., higher S.W.L.) it can be modified and applied accordingly if the required information is known.

**Funding:** This research received no external funding.

**Institutional Review Board Statement:** Not applicable.

**Informed Consent Statement:** Not applicable.

**Data Availability Statement:** Data available on request.

**Acknowledgments:** I would like to thank J.A. Jiménez for introducing me to the coastal vulnerability assessment methodology.

**Conflicts of Interest:** The author declares no conflict of interest.

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
