# Peer review of "Assessing Coastal Vulnerability to Storms: A Case Study on the Coast of Thrace, Greece"

_jmse, doi:10.3390/jmse11081490_

Round 1

Reviewer 1 Report

The paper entitled "Assessing the coastal vulnerability to storms: A case study at 2 the coast of Thrace, Greece" provides an interesting analysis of a current problem. While relevant, the paper needs a few improvements, in order to achieve its full potential. 

A series of comment can be found below:

- Providing visual information is important, but the images should have good quality. The quality of Figure 1 should be improved (300 dpi), and the same comment applies to all following figures. 

- The introduction is well structured and provides a good background for the study (the segment which includes examples of previous works is particularly relevant). However, it should be backed up with more references. For example, for the first 3 paragraphs, only 7 papers were referenced. The information provided is truthful and relevant, so it should be backed up by already published work. The introduction states the novelty of the work, which is much relevant.

- Figure 4, which schematizes the followed methodology is a great plus. It makes it much easier for the reader to fully to understand what was done during the study. However, it has very poor quality. It is extremely important to revise this.

- The methodology should also be more detailed. The following sections, which also entail the results, explain the methodology followed in more detail, but these aspects should be divided. Meaning: section 3 should just be methodology, and a fourth section should be created to display the attained results.

- Section 4, entitled Discussion, should be renamed to "Discussion and Conclusions".

- The topic of the paper is extremely interesting, and the Discussion indicates the ways the results could/should be improved and how they could be used by decision makers. However, it does feel like a slightly "simple" work. A more thorough discussion and conclusion section would upgrade the paper.

The quality of the English language is good, although some minor revisons should be made. As I am not a fluent English speaker, I did not mark any revision of the manuscript, but the MDPI Editing team surely will analyse this aspect. 

Author Response

I appreciate the reviewer’s interest in my article, and I would like to thank him/her for providing feedback on areas that require improvement. The revisions made in the manuscript are marked in green.

The paper entitled "Assessing the coastal vulnerability to storms: A case study at the coast of Thrace, Greece" provides an interesting analysis of a current problem. While relevant, the paper needs a few improvements, in order to achieve its full potential. A series of comment can be found below:

- Providing visual information is important, but the images should have good quality. The quality of Figure 1 should be improved (300 dpi), and the same comment applies to all following figures. 

The reviewer is right, I do not know why but the quality of the Figures in the pdf file is bad. In the revised manuscript, the quality of the Figures is improved.

- The introduction is well structured and provides a good background for the study (the segment which includes examples of previous works is particularly relevant). However, it should be backed up with more references. For example, for the first 3 paragraphs, only 7 papers were referenced. The information provided is truthful and relevant, so it should be backed up by already published work. The introduction states the novelty of the work, which is much relevant.

The introduction in the revised manuscript is modified and enhanced by six more relevant articles.

- Figure 4, which schematizes the followed methodology is a great plus. It makes it much easier for the reader to fully to understand what was done during the study. However, it has very poor quality. It is extremely important to revise this.

In the revised manuscript, the quality of the Figures is improved.

- The methodology should also be more detailed. The following sections, which also entail the results, explain the methodology followed in more detail, but these aspects should be divided. Meaning: section 3 should just be methodology, and a fourth section should be created to display the attained results.

I agree with the reviewer, it is better to have these sections clearly separated. In the revised manuscript, section 3 (Methodology) includes the methodology in detail, whereas section 4 (Results) presents solely the results.

- Section 4, entitled Discussion, should be renamed to "Discussion and Conclusions".

In the revised manuscript, section 5 is Discussion and section 6 is Conclusions.

- The topic of the paper is extremely interesting, and the Discussion indicates the ways the results could/should be improved and how they could be used by decision makers. However, it does feel like a slightly "simple" work. A more thorough discussion and conclusion section would upgrade the paper.

Section 5 (Discussion) and section 6 (Conclusions) are expanded in the revised manuscript.

The quality of the English language is good, although some minor revisions should be made.

The manuscript has been revised and corrected. It should be noted that these corrections are not highlighted in the revised manuscript for clarity.

Reviewer 2 Report

The author adapted the methodology described in Bosom and Jiménez (2011) to assess the vulnerability of a large coastline, and the coast of Thrace, Greece, was chosen as an application example. The author also claimed to provide a simple methodology for decision-makers to have a first glance at results and decide where and how to invest their limited resources. One of the main differences between the approach presented here and the one described in the previous study is that the author replaced the empirical formula with the numerical model SBEACH for storm-induced beach erosion estimation. My major comments are:

1.       What is the advantage or purpose of replacing the empirical formula with a numerical model? Since the author would like to provide decision-makers with a simple approach to have fast and reasonable results, the empirical approach seems more appropriate.

2.       Besides, a numerical model usually requires a lot of data to have decent results, which seems to violate the author's premises.

3.       The approach described by Bosom and Jiménez (2011) uses empirical formulae for both erosion and inundation. Their approach is based on the framework of "experiences," and the author replaced one empirical equation with a deterministic model. Does this attempt still fit in the framework of methodology? I would suggest the author rationalize his approach to convince readers.

4.       Would the safety requirement mentioned in eq(1) be appropriate? For instance, the retreat of 4.5 m in beaches 6 and 10 will cause different reactions from society. However, they all fall into the safe category according to the definition.

Several minor comments are as follows:

1.       The resolution of all figures is inadequate, and it isn't easy to read in detail.

2.       The color bar of Figure 2 has no unit.

3.       What is the spatial resolution of the DEM shown in Figure 2? It seems too coarse, and would it be appropriate for the input of SBEACH?

4.       How are all tables' physical parameters (e.g., length, width, height, etc.) defined? The max. or averaged?

5.       What is the height in all tables? It is defined as the beach height in Table 3, but the berm height is mentioned in L276.

6.       How is Δx defined? It would be better to illustrate in Figure 6 schematically.

7.       Would it be possible for beaches 1-4 and 11 not to be eroded by short-term storms (L198)? If so, what is the reason? It would be difficult to judge the results without information on waves.

8.       Please add explanations of R2% in eq(3) and Q in eq(7).

9.       Where does the accepted probability P=0.3 come from? Is there any reference literature supporting this number?

10.   Do the results of integrated coastal vulnerability shown in Figure 12 reflect the observations or experiences of the current situation?

11.   Although two run-up equations were presented in the manuscript for the estimation of inundation, the Stockdon eq. seems to provide no further information about the role while considering overtopping. Is it adequate to substitute the R2% calculated in Stockdon eq. to eq(7) for the corresponding overtopping discharge? Or is a more conservative value of R2% chosen for the following calculation of the overtopping discharge?

Author Response

I would like to thank the reviewer for pointing out that the rationale for selecting SBEACH was missing and for the other valuable comments. The revisions made in the manuscript are marked in green.

The author adapted the methodology described in Bosom and Jiménez (2011) to assess the vulnerability of a large coastline, and the coast of Thrace, Greece, was chosen as an application example. The author also claimed to provide a simple methodology for decision-makers to have a first glance at results and decide where and how to invest their limited resources. One of the main differences between the approach presented here and the one described in the previous study is that the author replaced the empirical formula with the numerical model SBEACH for storm-induced beach erosion estimation. My major comments are:

  1. What is the advantage or purpose of replacing the empirical formula with a numerical model? Since the author would like to provide decision-makers with a simple approach to have fast and reasonable results, the empirical approach seems more appropriate.

In L64-70 of the revised manuscript, the disadvantage of the empirical formula used in Bosom and Jimenez (2011) is presented.  The advantage of using SBEACH is stated in L189-198.

  1. Besides, a numerical model usually requires a lot of data to have decent results, which seems to violate the author's premises.

The reviewer is right, an empirical formula is usually more straightforward than a numerical model. However, SBEACH is not so time consuming neither in terms of setting it up nor in simulating the erosion process. L197-198

  1. The approach described by Bosom and Jiménez (2011) uses empirical formulae for both erosion and inundation. Their approach is based on the framework of "experiences," and the author replaced one empirical equation with a deterministic model. Does this attempt still fit in the framework of methodology? I would suggest the author rationalize his approach to convince readers.

The rationale for selecting SBEACH is that the empirical formula used to estimate the erosion in Bosom and Jiménez (2011) was derived from numerical simulations of SBEACH along the Catalonian coast (Mendoza and Jimenez, 2006). Hence, their empirical formula could not be used in other beaches (see also the response in comment 1). Since SBEACH is rather easy to use, it was selected instead of another empirical formula.

Mendoza, E.T.; Jiménez J.A. Storm-induced beach erosion potential on the Catalonian coast. J. of Coast. Res. 2006, 81-88.

  1. Would the safety requirement mentioned in eq(1) be appropriate? For instance, the retreat of 4.5 m in beaches 6 and 10 will cause different reactions from society. However, they all fall into the safe category according to the definition.

In this work, the safety requirement is selected in order to leave sufficient space (10 m was assumed) for the machinery work to restore the storm-induced damages. Different values of the safety requirement will lead to different vulnerability assessments as the reviewer correctly points out. For this article, the used safety requirement value is just a suggestion to present the methodology. Obviously, different values should be used by taking into consideration the particularities of each region. The same also applies to inundation.

Several minor comments are as follows:

  1. The resolution of all figures is inadequate, and it isn't easy to read in detail.

The reviewer is right, I do not know why but the quality of the Figures in the pdf file is bad. In the revised manuscript, the quality of the Figures is improved.

  1. The color bar of Figure 2 has no unit.

This is corrected in the revised manuscript.

  1. What is the spatial resolution of the DEM shown in Figure 2? It seems too coarse, and would it be appropriate for the input of SBEACH?

The spatial resolution of the Hellenic Military Geographical Service is 5 m. This is considered adequate for the present analysis. Limitations about the beach dimensions are also discussed in (L390-393). Note that in the revised manuscript (L120-122) it is stated that Figure 2 shows the bathymetry of the studied area and the location of the 4 stations. Information about the beach dimensions was obtained from the Hellenic Military Geographical Service (L132-134) not from the bathymetry of Figure 2.

  1. How are all tables' physical parameters (e.g., length, width, height, etc.) defined? The max. or averaged?

The tables’ physical parameters were averaged. See also L390-393 in the revised manuscript.

  1. What is the height in all tables? It is defined as the beach height in Table 3, but the berm height is mentioned in L276.

            The height is defined as beach height in the revised manuscript to avoid confusions.

  1. How is Δx defined? It would be better to illustrate in Figure 6 schematically.

The beach retreat is illustrated in Figure 5 of the revised manuscript.

  1. Would it be possible for beaches 1-4 and 11 not to be eroded by short-term storms (L198)? If so, what is the reason? It would be difficult to judge the results without information on waves.

Beaches 1-4 and 11 were found to be not eroded at all even if the waves attacking them were higher than the eroded beaches. The reason for this is stated in the revised manuscript in L305-306.

  1. Please add explanations of R2% in eq(3) and Q in eq(7).

In the revised manuscript, the explanations of both variables are added in L228-231 and L262-263, respectively.

  1. Where does the accepted probability P=0.3 come from? Is there any reference literature supporting this number?

This is a recommendation of the Greek Ministry of Public Works (L370-374 in the revised manuscript). It should be noted that the selected values of the probability and period of concern in this article is to demonstrate the methodology. It is encouraged to use values depending on the recommendations of the decision-makers of each region.

  1. Do the results of integrated coastal vulnerability shown in Figure 12 reflect the observations or experiences of the current situation?

Unfortunately, I cannot answer for all the examined beaches since I cannot find any scientific reference about that. However, I know that there are problems of inundation and erosion along the beach of Alexandroupolis (Beach 10) and the coastline of Rodopi (Beaches 4, 5 and 6) after searching the Greek newspapers.

  1. Although two run-up equations were presented in the manuscript for the estimation of inundation, the Stockdon eq. seems to provide no further information about the role while considering overtopping. Is it adequate to substitute the R2% calculated in Stockdon eq. to eq(7) for the corresponding overtopping discharge? Or is a more conservative value of R2% chosen for the following calculation of the overtopping discharge?

In L351-352 of the revised manuscript, the explanation of why only the Reis equation was used is given.

Reviewer 3 Report

The authors studied the coastal vulnerability to storms at the coast of Thrace, Greece. The study fits well with the scope of the journal. My detailed comments are as follows:

1. How interactions between astronomical tides and storm affect the vulnerability?

2. Details of the model framework are missing, such as how the wave run-up time-series are calculated, as well as the erosion & inundation.

3. Line 187, the authors are recommended to state the disadvantages in details instead of simply providing a reference.

4. Why Stockdon and Reis equations were selected for this study? The pronounced difference between results from the Stockdon and Reis equation could due to the misuse of the models.

Park, Hyoungsu, and Daniel T. Cox. "Empirical wave run-up formula for wave, storm surge and berm width." Coastal Engineering 115 (2016): 67-78.

6. For Stockdon equation, what is the parameter range the equation was developed? Does it work better with high energy or low energy events? Also, the equation was developed using data sets from US East and West Coast beaches, the performance for the study needs to be evaluated.

 7. 3-h hindcasts of the WAM model for ten years between 1995-2004 were used in this study. As the authors point out, the intensity of the storms could change due to climate change, were more recent data available? Also, how would the conclusion be affected by climate change under future scenarios?

There are a lot of typos and grammar errors spreading through the manuscript, and a thorough proofread is recommended.

Author Response

I would like to thank the reviewer for his/her careful suggestions regarding the article. The revisions made in the manuscript are marked in green.

The authors studied the coastal vulnerability to storms at the coast of Thrace, Greece. The study fits well with the scope of the journal. My detailed comments are as follows:

  1. How interactions between astronomical tides and storm affect the vulnerability?

            In L225-228 of the revised manuscript, this is explained.

  1. Details of the model framework are missing, such as how the wave run-up time-series are calculated, as well as the erosion & inundation.

In the revised manuscript, section 3 (Methodology) includes the methodology of how the erosion and inundation were calculated in detail.

  1. Line 187, the authors are recommended to state the disadvantages in detail instead of simply providing a reference.

In L189-196 and L401-405 of the revised manuscript, the disadvantages are stated.

  1. Why Stockdon and Reis equations were selected for this study? The pronounced difference between results from the Stockdon and Reis equation could due to the misuse of the models. Park, Hyoungsu, and Daniel T. Cox. "Empirical wave run-up formula for wave, storm surge and berm width." Coastal Engineering 115 (2016): 67-78.

Stockdon equation is a well-established formula for estimating the wave run-up because it is derived from ten field experiments, and this is the reason why it was selected first. The article of Laudier et al. (2011) that observed underestimations of Stockdon eq. and better accuracy of the Reis eq., triggered the idea of analyzing the results of both formulas (L236-241).

  1. For Stockdon equation, what is the parameter range the equation was developed? Does it work better with high energy or low energy events? Also, the equation was developed using data sets from US East and West Coast beaches, the performance for the study needs to be evaluated.

As the reviewer states, the Stockdon eq. was developed using data sets from US East and West Coast beaches. This means that apart from storm waves, powerful swells also hit these coastlines. This is in contrast with the beaches of Thrace where the wave climate is only storm waves (short wavelengths and high wave heights). However, since Stockdon eq. is an established formula to estimate the wave run-up, it was selected to compare its results in this region.

  1. 3-h hindcasts of the WAM model for ten years between 1995-2004 were used in this study. As the authors point out, the intensity of the storms could change due to climate change, were more recent data available? Also, how would the conclusion be affected by climate change under future scenarios?

Unfortunately, more recent data was not available. However, the used wave data was compared by Galiatsatou and Prinos in 2011, and its accuracy was acceptable (L124-127 of the revised manuscript). Without having information about the climate change scenarios, no conclusions about the results can be drawn of the revised manuscript (L430-332.

There are a lot of typos and grammar errors spreading through the manuscript, and a thorough proofread is recommended.

            A thorough proofread was performed following the reviewer recommendation.

Reviewer 4 Report

In this MS, the author tried to assess the the coastal vulnerability by considering  erosion and inundation phenomena. First, the erosion is computed using the numerical model SBEACH, whereas the inundation is estimated using two different empirical equations for comparison. He mentioned that the integration of the vulnerabilities of both storm-induced impacts associated to the same return period permits the identification of the most vulnerable areas. 

The findings could be interesting to the readers. However, It needs more improvements: 

Abstract: 1. Please elaborate SBEACH in first mention. 

2. Could not find the significant results- please include those. 

3. How can this data be useful for the area? please mention.

Introduction: 

1. Please delete figure from the introduction

2. Move para five to top ( ist para)- to introduce the topic.

3. correct line 56

4. In para 4, please mention the established ways of vulnerability assessment and their advantages and disadvantages

Methods: Please justify why did you use the SBEACH model?

Please separate the results from the methods. It is hard comprehend in the current stage.

Discussion: Please arrange the discussion based on the results. 

Conclusion: I missed it. Please add. 

Thank you

Moderate editing is needed.

Author Response

I would like to thank the reviewer for his/her careful suggestions regarding the article. The revisions made in the manuscript are marked in green.

In this MS, the author tried to assess the coastal vulnerability by considering erosion and inundation phenomena. First, the erosion is computed using the numerical model SBEACH, whereas the inundation is estimated using two different empirical equations for comparison. He mentioned that the integration of the vulnerabilities of both storm-induced impacts associated to the same return period permits the identification of the most vulnerable areas.  The findings could be interesting to the readers. However, it needs more improvements:

Abstract: 1. Please elaborate SBEACH in first mention.

In the revised manuscript, SBEACH is elaborated in the abstract (L11) and in the first mention of the main text (L47-48).

  1. Could not find the significant results- please include those.

            L14-18 in the revised manuscript are added.

  1. How can this data be useful for the area? please mention.

            L18-19 in the revised manuscript are added.

Introduction:

  1. Please delete figure from the introduction

In the revised manuscript, Figure 1 is deleted from the introduction, and it is moved to section 2 (Study coast and data).

  1. Move para five to top ( ist para)- to introduce the topic.

In the revised manuscript, paragraph 5 (of the first submission) is moved to the top of the introduction, following the suggestion of the reviewer.

  1. correct line 56

L59 in the revised manuscript is rephrased.

  1. In para 4, please mention the established ways of vulnerability assessment and their advantages and disadvantages.

Paragraph 4 (of the first submission) is modified in the revised manuscript (L59-77).

Methods: Please justify why did you use the SBEACH model?

The rationale for selecting SBEACH is that the empirical formula used to estimate the erosion in Bosom and Jiménez (2011) was derived from numerical simulations of SBEACH along the Catalonian coast (Mendoza and Jimenez, 2006). Hence, their empirical formula could not be used in other beaches (L64-70). Since SBEACH is rather easy to use, it was selected instead of another empirical formula (L189-198).

Mendoza, E.T.; Jiménez J.A. Storm-induced beach erosion potential on the Catalonian coast. J. of Coast. Res. 2006, 81-88.

Please separate the results from the methods. It is hard to comprehend in the current stage.

I agree with the reviewer, it is better to have these sections clearly separated. In the revised manuscript, section 3 (Methodology) includes the methodology in detail, whereas section 4 (Results) presents solely the results.

Discussion: Please arrange the discussion based on the results.

Section 5 (Discussion) is arranged based on the results and further expanded.

Conclusion: I missed it. Please add.

Section 6 (Conclusions) is added.

Thank you

I would like to thank the reviewer for his/her suggestions regarding the article.

Moderate editing is needed.

            The manuscript has been revised and corrected.

Round 2

Reviewer 1 Report

The authors has successfully adjusted the submitted paper, according to the suggestions made. The overall quite has improved, and the new structure of the document provides an easier reading and analysis experience - when compared to the originally submitted version.

However: the quality of the images is still quite low. Figures 6, 7, 9 and 10 are very hard to read, for example. I believe this should be changed, before the paper gets published. It is the only matter I have to point out.

No comments.

Author Response

The author has successfully adjusted the submitted paper, according to the suggestions made. The overall quite has improved, and the new structure of the document provides an easier reading and analysis experience - when compared to the originally submitted version.

However: the quality of the images is still quite low. Figures 6, 7, 9 and 10 are very hard to read, for example. I believe this should be changed, before the paper gets published. It is the only matter I have to point out.

I realized after the two submissions that the problem with the image resolution derives from the automatic formation of the pdf file. On both previous submitted word files, the resolution is good. To deal with it, I have uploaded all images in good quality and since the production of the manuscript will be based on them, I am not expecting any problems. I also informed the assistant editor about this issue.

Reviewer 2 Report

There are still some unclear parts in the revised manuscript. Here are some comments:

1.       All figures are still blurred and difficult to read; some have issues. For example, the color of the ocean is white in the top-left corner but black in the top-right corner in Figure 1; the marks indicating four stations are difficult to be recognized in Figure 2. A proper legend indicating stations might be helpful.

2.       The author explains the selection of the Reis equation in L351-352 to be on the safe side when compared with the run-up results of the Stockdon eq. Afterward, an empirical eq(7) was adopted to calculate overtopping discharge. Finally, the exceeding height of two meters was determined, corresponding to the two-meter height in eq(6). If this way determines the two-meter height in eq(6), the Stockdon eq. seems unnecessary here. Furthermore, according to L352-354, the overtopping discharge of 150 lt/s is the main criterion for determination of the exceeding height. Would it be possible to have a more conservative exceeding height using another empirical equation of overtopping discharge suited for the Stockdon eq.? The logical order in this part seems to be ambiguous. Besides, I didn’t find eq(7) in the provided reference [36].

3.       L300-306 describe the erosion results and explain why beaches 1-4 and 11 are not eroded compared with other beaches, which claimed to be located in the sheltered region. The explanation is mainly due to the beach slope, which dissipates wave energy more. However, the bathymetry of all the study areas shown in Figure 2 seems mild. Due to wave refraction and diffraction, wouldn’t beaches 1-4 and 11 also be in the sheltered areas by Thasos and Samothraki? For instance, wouldn’t beach 1 be in the sheltered region of Thasos if waves come from the SW direction?

Author Response

There are still some unclear parts in the revised manuscript. Here are some comments:

  1. All figures are still blurred and difficult to read; some have issues. For example, the color of the ocean is white in the top-left corner but black in the top-right corner in Figure 1; the marks indicating four stations are difficult to be recognized in Figure 2. A proper legend indicating stations might be helpful.

The problem with the image resolution derived from the automatic formation of the pdf file. I am expecting that it will be improved in the revised manuscript, because I reuploaded all figures in good quality separately. I also informed the assistant editor about that. Furthermore, Figures 1 and 2 are modified according to the sharp observations of the reviewer.

  1. The author explains the selection of the Reis equation in L351-352 to be on the safe side when compared with the run-up results of the Stockdon eq. Afterward, an empirical eq(7) was adopted to calculate overtopping discharge. Finally, the exceeding height of two meters was determined, corresponding to the two-meter height in eq(6). If this way determines the two-meter height in eq(6), the Stockdon eq. seems unnecessary here. Furthermore, according to L352-354, the overtopping discharge of 150 lt/s is the main criterion for determination of the exceeding height. Would it be possible to have a more conservative exceeding height using another empirical equation of overtopping discharge suited for the Stockdon eq.? The logical order in this part seems to be ambiguous. Besides, I didn’t find eq(7) in the provided reference [36].

Indeed, this part was a bit ambiguous after the modifications of sections 3 and 4 during the first review round. To clarify it in the revised manuscript, the variable Z is added in equation 6, because its value is not known a priori (L257-269). The selection of its value is presented in section 4.2 (L359) as an example for the specific conditions of the studied coastline. It should be noted that other equations to estimate the overtopping discharge could be used. In this study, the equation in Laudier et al. 2011 was used to demonstrate the methodology of selecting the variable Z. The reference of Eq. (7) is corrected in the revised manuscript (L265) (Eq. (9) in Laudier, N.A.; Thornton, E.B.; MacMahan, J. Measured and modeled wave overtopping on a natural beach’, Coast. Eng. 2011, 58, 815-825).

  1. L300-306 describe the erosion results and explain why beaches 1-4 and 11 are not eroded compared with other beaches, which claimed to be located in the sheltered region. The explanation is mainly due to the beach slope, which dissipates wave energy more. However, the bathymetry of all the study areas shown in Figure 2 seems mild. Due to wave refraction and diffraction, wouldn’t beaches 1-4 and 11 also be in the sheltered areas by Thasos and Samothraki? For instance, wouldn’t beach 1 be in the sheltered region of Thasos if waves come from the SW direction?

The slope of each beach is presented in the penultimate column of Table 2. The maximum slope of beaches 1-4 and 11 is 0.02 (beach 4), whereas the minimum slope of beaches 5-10 is 0.032 (beach 9). Beaches 1-4 and 11 are sheltered due to Thasos and Samothraki but they are more exposed than the other studied beaches. (The word “more” is added in L308 and the reference to Table 1 is added in L310 to clarify these issues).

I would like to thank the reviewer for providing more feedback on areas that require improvement. The revisions made in the manuscript are marked in green.

Reviewer 4 Report

Please correct the citations in some sentences like in line 59: 'Studies that considered both the erosion and inundation impacts to estimate the integrated coastal vulnerability were first presented in [12,3,13,14].' Please see some other papers published in MDPI journals. We can not write .... presented in ....and then within brackets ...

Minor English editing requires. 

Author Response

Please correct the citations in some sentences like in line 59: 'Studies that considered both the erosion and inundation impacts to estimate the integrated coastal vulnerability were first presented in [12,3,13,14].' Please see some other papers published in MDPI journals. We can not write .... presented in ....and then within brackets ...

The citations were corrected throughout the manuscript following the suggestion of the reviewer.